# *Ccr6* Deficiency Attenuates Spontaneous Chronic Colitis in Winnie

**Ranmali Ranasinghe [1], Ruchira Fernando [2], Agampodi Promoda Perera [1] , Madhur Shastri [1], Waheedha Basheer [1], Paul Scowen [3], Terry Pinfold [3] and Rajaraman Eri [1,*]**

[1] School of Health Sciences, College of Health and Medicine, University of Tasmania, Launceston 7250, Tasmania, Australia; hewausaramba.ranasinghe@utas.edu.au (R.R.); agampodi.perera@utas.edu.au (A.P.P.); madhur.shastri@utas.edu.au (M.S.); waheedha.basheer@utas.edu.au (W.B.)

[2] Department of Histopathology, Launceston General Hospital, Launceston 7250, Tasmania, Australia; ruchira.fernando@ths.tas.gov.au

[3] School of Medicine, University of Tasmania, Hobart 7000, Tasmania, Australia; paul.scowen@utas.edu.au (P.S.); terry.pinfold@utas.edu.au (T.P.)

\* Correspondence: rderi@utas.edu.au

**Abstract:** Background: The immune-modulator behaviour of the CCR6/CCL20 axis in multi -system pathophysiology and molecular signalling was investigated at two clinically significant time points, using a *Ccr6*—deficient mouse model of spontaneous colitis. Methods:Four groups of mice, (C57BL/6J, *Ccr6*$^{-/-}$ of C57BL/6J, Winnie × *Ccr6*$^{-/-}$ and Winnie) were utilized and (I) colonic clinical parameters (2) histology of colon, spleen, kidney and liver (3) T and B lymphocyte distribution in the spleen and MLN by flowcytometry (5) colonic CCL20, phosphorylated PI3K and phosphorylated Akt expression by immunohistochemistry and (6) colonic cytokine expression by RT-PCR were evaluated. Results: CCR6 deficiency was shown to attenuate inflammation in the spleen, liver and gut while renal histology remained unaffected. Marked focal lobular inflammation with reactive nuclear features were observed in hepatocytes and a significant neutrophil infiltration in red pulp with extra medullary hemopoiesis in the spleen existed in Winnie. These changes were considerably reduced in Winnie × *Ccr6*$^{-/-}$ with elevated goblet cell numbers and mucus production in the colonic epithelium. Conclusions: Results indicate that *Ccr6*-deficiency in the colitis model contributes towards resolution of disease. Our findings demonstrate an intricate networking role for CCR6 in immune activation, which is downregulated by *Ccr6* deficiency, and could provide newer clinical therapies in colitis.

**Keywords:** CCR6; colitis; Winnie; multi-system pathophysiology; molecular signalling; inflammation; immunity

---

## 1. Introduction

Inflammatory bowel disease (IBD) is a globalised immune compromised disease complex having two concordant phenotypes; Crohn's disease (CD) and ulcerative colitis (UC) which manifest in the human gut [1]. These diseases are characterized by chronic intestinal inflammation [2]. IBD causes severe morbidity in young adults and costs the health sector a three-fold increase in expenditure compared to non-IBD controls [3,4]. The disease itself places a heavy burden on the patients by causing psychological distress such as annihilating career objectives, instigating social stigma and deteriorating their quality of life [5]. It produces a series of relapses and remissions with colectomy as the final treatment option. Prolonged disease leads to a high risk of developing colitis-associated colorectal cancer (CAC) [6].

Chemokines are small protein molecules which act as chemoattractant cytokines and consist of two cohorts, the receptors and ligands [7].CC-chemokine receptor 6 (CCR6) and its sole ligand,

CC-chemokine ligand 20 (CCL20) are an important chemokine pair which performs a pivotal role in causing inflammation in the gut [8].

CCL20 is produced in copious amounts by the gut epithelium in response to infectious microbial penetration which attracts the immune cells bearing CCR6 to the site of infection [7]. CCR6-CCL20 axis fundamentally aids leukocyte homeostasis in the intestines and the CCR6-deficient mouse model is used to understand how the immune mechanisms work towards restoring T and B lymphocyte balance in the steady state as seen in a healthy person [9].

It is because the CCR6-CCL20 axis is critically involved in determining the immune cell imbalance of TH17 and Treg lymphocyte cohorts, initiating chronic colitis in the gut [10]. Not only colitis, but many other diseases relating to multi-organ pathologies are also influenced by the CCR6-CCL20 immune axis. Several autoimmune diseases such as experimental autoimmune encephalitis (EAE), rheumatoid arthritis (RA) and psoriasis have been shown to display reduced inflammation when CCR6 was inhibited in pre-clinical and clinical disease models [11]. Sixteen tractable CCR6 inhibitors have been introduced as a potential therapeutic option in those diseases [11]. In addition to having pathology within the gastro-intestinal tract, IBD patients also exhibit secondary organ pathologies termed as extraintestinal manifestations [12]. The spleen, liver and kidneys which are immune-related organs of secondary lymphoid origin were examined as they cause the systemic diseases—hepatitis and glomerular nephritis, where CCR6 plays a crucial role in initiating inflammation [13–16].

This led us to investigate the phenotype of the *Ccr6*-deficient Winnie mouse model and the influence of CCR6-CCL20 in multiple organs during the manifestation of colitis and its contribution towards initial molecular signalling. A plethora of therapies have been tried to date in the treatment of inflammatory bowel disease (IBD), however not, CCR6 -inhibition [11].

This study aims at elucidating the effects of CCR6-deficiency involving multi-systems in an inflammatory setting of chronic spontaneous colitis. Chronic colitis in Winnie is caused by a primary epithelial cell defect activated by a point mutation in the *Muc2* gene resulting in aberrant mucin-2 biosynthesis [17]. It leads to endoplasmic reticulum stress in intestinal goblet cells and reduced secretion of mucus [18]. Winnie mice display immune responses by a threefold increase of immune cells in the colon compared to WT mice with a significant increase in mucosal cytokine secretion in the gut and produce disease like that of human UC [19]. Winnie, a time-tested and proven model of murine spontaneous colitis which closely resembles human ulcerative colitis (UC) was used in which the *Ccr6* gene was knocked out [17].

Winnie mice display symptoms of diarrhoea, ulcerations, rectal bleeding and pain at different stages of colitis synonymous with human disease. Exhaustive studies done in Winnie have proven it to be one of the best available murine models to study human chronic colitis and its pathogenesis [19]. Chronic gut inflammation is known to take a hold by 6 weeks of age in Winnie and manifests into chronic colitis by 16 weeks of age [17]. Therefore, this study proves to be highly clinically relevant in most of its aspects.

The anticipated benefits that justify our study are the existing compelling evidence that CCR6 has multiple immune functions in many organ systems of humans and is considered a valuable therapeutic target [20]. From preclinical studies, it is hoped that the immune cell triggers or pathways that arise from CCR6 activation could be identified [8]. These results could then be translated into clinical trials which target the neutralization of CCR6 or its associated cytokines/products/mediators in human colitis. No such clinical trials have been performed with the existing CCR6 inhibitors up to date in IBD, which is also considered by some as an autoimmune disease [11].

The importance of CCL20, phosphorylated PI3K and phosphorylated Akt are that they initiate the immune activation of CCR6 and lead to varied immune pathways which involve colitis and general metabolism [8,21]. The aim of this study was to investigate the influence of CCR6 deficiency in multiorgan inflammation in Winnie × *Ccr6*$^{-/-}$ and our results show a considerable reduction in inflammation in the colon, spleen and liver, evidenced by the disease activity index (DAI) and constructed histological scoring schemes.

In our study we found that deletion of the *Ccr6* gene in mice expressing spontaneous colitis displayed increased resistance to colonic inflammation. Colonic inflammation was limited to the proximal colonic segment and an inflammation-free mid to distal colon has been observed by our group [9]. We determined some important clinical, histological and immunological parameters which delineates the impact of CCR6—deficiency in terms of its potential to become a proactive drug target. These findings are expected to open avenues to a more detailed examination of CCR6 immunobiology and physiology to evaluate the impact of CCR6—inhibition as a novel treatment option in IBD.

## 2. Materials and Methods

### 2.1. Animals

All mice experiments were conducted under the Ethics permit number A17451 of the Animal Ethics Committee of the University of Tasmania in accordance with the Australian Code of Practice for Care and Use of Animals for Scientific Purposes (8th Edition, 2013). Mice were housed in a temperature-controlled PC2 animal facility with a 12-hour day/night light cycle. Individual body weights were maintained daily over an initial acclimation period of 7 days. All mice were given radiation-sterilised rodent feed and autoclaved tap water for drinking ad libitum during experiments. All efforts were made to minimize animals' suffering and to reduce the number of animals used. Mice utilised (Table 1), were in the age group of 8–22 weeks.

**Table 1.** Experimental strains of mice utilised with a description of age, sex, phenotype, sample size and their role in the research project. The mice were utilised in two batches, at 8 weeks and 16–22 weeks for separate pre-clinical assessment.

| Animal Strain | Age | Sex | Phenotype | Number | Description |
|---|---|---|---|---|---|
| C57BL/6J | 8–22 weeks | M/F | Wild type (WT) | 16 | Healthy Control |
| *Ccr6*$^{-/-}$ | 8–22 weeks | M/F | Targeted knockout | 16 | Negative Control |
| Winnie × *Ccr6*$^{-/-}$ | 8–22 weeks | M/F | Targeted knockout | 16 | Experimental Model |
| Winnie | 8–22 weeks | M/F | *Muc2* mutation | 16 | Positive Control |

A *Ccr6* deficient C57BL/6J (WT) male was mated with a WT female and the fertilized embryos were harvested and implanted to a Swiss Webster surrogate female mouse. The littermates consisted of males and females which were heterozygous for *Ccr6*. A bisexual pair of *Ccr6* heterozygotes were mated to obtain a *Ccr6*$^{-/-}$ male which was crossed with a Winnie female, double recessive for *Ccr6*$^{-/-}$. The *Ccr6*$^{-/-}$ male was homozygous dominant for *Muc2* and the Winnie female was homozygous dominant for *Ccr6*. The littermates from this pair were heterozygous, Winnie × *Ccr6*$^{+/-}$, which were mated to obtain a double homozygous Winnie × *Ccr6*$^{-/-}$. The breeding of animals was done at the Cambridge Farm Facility, affiliated to the University of Tasmania.

### 2.2. Phenotypic Assessment

The phenotype was assessed using clinical parameters of each group of mice. The body weight was recorded prior to culling. The mice were culled using carbon dioxide euthanasia and were dissected under sterile conditions. The large intestine was separated at the ileo-caecal junction and the anus and placed on a non-absorbent surface. The length of the colon was measured from the ileo-caecal junction to the anus and was recorded. The whole colon was cut open along the longitudinal axis, faecal matter removed, and weight was recorded.

### 2.3. Morphological Assessment of Other Organs

The spleen, kidneys and liver were harvested and weight of each was recorded. Both the kidneys were weighed together and general morphology such as size, colour, shape and texture were observed.

## 2.4. Flowcytometry

Mouse spleen was stored in ice-cold x1 phosphate buffered saline (PBS; pH 7.4- Dulbecco's, $10 \times$ L, Gibco, Catalogue No. 21300025, Life Technologies Pty Ltd, Victoria, Australia) on ice after which it was processed for analysis by flowcytometry. Mesenteric lymph nodes were harvested and stored in ice-cold RPMI 1640 culture medium (Catalogue No. 11875, Life Technologies Pty Ltd.). Spleen and mesenteric lymph nodes were macerated using the end of a syringe plunger with ice-cold PBS and passed through a 70 µ corning cell strainer (Catalogue No. 08-771-2, Thermo Fisher Scientific Pty Ltd, Victoria, Australia) to prepare a homogenous suspension which was centrifuged at 500 g for 10 min at 4 °C (Allegra X15R Beckman Coulter, Indianapolis, IN, USA) to obtain a cell pellet. The RBC were removed by adding RBC lysis buffer (Catalogue No. 420301, Australian Biosearch Pty Ltd, Wangarra, WA, Australia). Ice-cold PBS was added to inactivate the RBC lysis buffer and centrifuged at 500 g for 7 min at 4 °C. FACS buffer was added to the RBC-free splenocyte samples and kept on ice. The number of cells were enumerated using trypan blue and $2 \times 10^7$ cells per mL was adjusted by adding FACS buffer. Sample was centrifuged at 500 g for 7 min in a microcentrifuge to obtain a cell pellet. The supernatant was discarded, and fluorescent conjugated antibody (CD4- BV421, CD8a- AF488, CD19-PE) was added and vortexed briefly to mix and incubated on ice in the dark for 30 min. The cells were centrifuged in PBS at 500 g for 7 min, suspended in FACS buffer and flow cytometric data was obtained using a BD FACS CANTO™ flow cytometer (BD Biosciences, La Jolla, CA, USA) and analysed using FCS Express version 6.06.0014 (De Novo Software, Pasadena, CA, USA) for windows. Mesenteric lymph node suspensions were processed the same way except for the addition of RBC lysis buffer.

## 2.5. Histology

One half of the colon cut length wise was rolled into a swiss roll, liver, spleen and kidney and preserved in 10% neutral buffered formalin and transferred into 70% ethanol after 1 day. After chemical processing, were embedded in paraffin and sections cut to a thickness of 5 microns by a rotary microtome and were stained with Gill's haematoxylin and eosin (H&E) (Catalogue No. 72611, Life Technologies Pty Ltd.). The H&E stained sections were imaged using a DP72 microscope (Olympus, New South Wales, Australia). The tissue sections from all animals were scored in a blinded manner after grading for colitis by the histopathologist involved in this study at the Launceston General Hospital. Cell counting was performed manually in 10 random microscopic fields at magnifications of ×200.

## 2.6. Alcian Blue Staining of Mucus Producing Goblet Cells

Deparaffinized slides were rehydrated in a series of alcohol. Stained with Alcian blue pH 2.5 (Catalogue number: B8438, Sigma-Aldrich, NSW, Australia) for 30 min, washed in running water for 2 min, rinsed in distilled water, counterstained with 0.1% nuclear fast red (Catalogue number: 60700, Sigma-Aldrich, NSW, Australia) for 5 min, washed and dehydrated in alcohol, cleared in xylene and mounted. Acid mucins stained blue and nuclei were stained in pink.

## 2.7. Immunohistochemistry

Paraffin-embedded colon tissue sections were dewaxed, processed in xylene and rehydrated in a series of ethanol. Antigen retrieval was carried out, hydrogen peroxide incubation was performed followed by the protein blocking using the Rabbit specific HRP/DAB (ABC) detection IHC kit (Catalogue No. 64261, Abcam, Cambridge, UK) following the manufacturer's protocol. Slides were washed under running water and rinsed in PBS (Dulbecco's, Invitrogen, Victoria, Australia) after each treatment. Primary antibody diluted 1:500 was added and incubated for 1 hour in the dark inside a humidified chamber at room temperature (RT) followed by secondary antibody for 30 min. DAB chromogen + substrate buffer was added to the tissue sections for 10 min and washed in PBS and counterstained

with strong liquid Mayer's haematoxylin for 30 s, dipped in ammonia water and dehydrated in ethanol and xylene, cover slipped after adding a mounting medium. Percentage expression of intensity was calculated by the formula; optical density (OD) = log (maximum intensity/mean intensity), where max intensity = 255 for 8-bit images using Fiji Image J Version 1.64 software for windows 7 (Scijava Software Inc, Suwa Rewa, Fiji).

### 2.8. Real Time Polymerase Chain Reaction (RT-PCR)

Colon tissue less than 30 mg in weight was crushed and homogenized using a plastic disposable tip and RNA was extracted usingan RNeasy mini kit (50) (Catalogue No. 74104Qiagen, Victoria, Australia) following the manufacturer's protocol. 100 μl of RNA was eluted using a spin column and purity of eluted RNA was determined using the ratio of absorbance at 260 nm/280 nm using an Eppendorf Bio Photometer (Thermo Fisher Scientific, Victoria, Australia). Samples above 2 were utilised for RT-PCR. Complementary DNA (cDNA) was synthesized from RNA samples using a High Capacity cDNA Reverse Transcription Kit (Catalogue No. 4368814, Thermo Fisher Scientific Australia Pty Ltd.) using the reaction conditions suggested by the manufacturer. 100 ng of cDNA from each sample was added to a PCR reaction including TaqMan Fast Master Mix (Catalogue No. 4444557, Thermo Fisher Scientific Australia Pty Ltd.) and a single gene-specific primer set supplied in the kit. Fast SYBR™ Green Master Mix (Catalogue No. 4385612) was used with the cDNA template, forward and reverse DNA primers in a reaction volume of 20 μl in pre-determined volumes as per manufacturer's protocol. The PCR plate was run on Applied Biosystems StepOnePlus™ Real-Time PCR systems (version 2.2.3). The samples were run in duplicate and an average CT (Cycle Threshold) value was calculated. The CT value for the housekeeping gene *Gapdh* was subtracted from the CT value of the gene under investigation. Gene expression was quantified using the comparative (ΔΔCT) method where the threshold cycle (CT) for each gene was normalised to reference gene *Gapdh*. Relative gene expression in the animals was presented as $2^{-\Delta CT}$.

### 2.9. Statistical Data Analysis

The data was analysed using Graph Pad Prism version 7.0 (Graph Pad Software Inc., San Diego, CA, USA). All data were expressed as mean ± standard error of the mean (SEM). The data were evaluated with One-way analysis of variance (ANOVA) and comparisons between groups were analysed using Tukey's multiple comparison tests. The data were considered significant when $p \leq 0.05$ (*), $p \leq 0.01$ (**), $p \leq 0.001$ (***) and $p \leq 0.0001$ (****).

## 3. Results

### 3.1. Clinical Parameters Indicate that Ccr6-Deficiency Lessens Inflammation in the Colon

#### 3.1.1. Clinical Parameters

As part of investigating the clinical parameters, the colon length, colon weight, body weight and colon weight/body weight ratio in the four genotypes were evaluated. Colon length in WT remained in similar proportions to the other three genotypes at 8 weeks of age (WT: 8.4 ± 0.14 cm, $Ccr6^{-/-}$: 8.4 ± 0.25 cm, Winnie × $Ccr6^{-/-}$: 8.2 ± 0.25 cm, Winnie: 8.6 ± 0.27 cm).

At 16–22 weeks (WT: 8.0 ±0.33 cm, $Ccr6^{-/-}$: 8.5 ± 0.6 cm, Winnie x $Ccr6^{-/-}$: 9.0 ± 0.46 cm, Winnie: 10 ± 0.74 cm), the colon length was longest in Winnie and it was shortest in the WT. As seen in Figure 1, a significant increase in colon length exists at the 16–22 weeks - time point between WT and Winnie (*** $p < 0.001$) as well as WT and Winnie × $Ccr6^{-/-}$(** $p \leq 0.01$) but not in the colon length at 8 weeks of age. The wet colon weight was measured using freshly removed colons, after emptying the luminal contents. At 8 weeks, (WT: 0.1± 0.02 g, $Ccr6^{-/-}$: 0.15 ± 0.02 g, Winnie × $Ccr6^{-/-}$: 0.29 ± 0.07 g, Winnie: 0.36 ± 0.03 g) and at 16–22 weeks (WT: 0.1 ± 0.03 g, $Ccr6^{-/-}$: 0.63 ± 0.01 g, Winnie × $Ccr6^{-/-}$: 0.5 ± 0.09 g, Winnie: 0.6 ± 0.09 g) a significant increase (**** $p < 0.0001$) in the colon weight was observed in Winnie

$\times$ *Ccr6*$^{-/-}$ compared to the WT. The colon weight of Winnie $\times$ *Ccr6*$^{-/-}$ remained significantly lower (* $p < 0.05$) than that of Winnie, at 8 weeks but not at the second time point (Figure 1). As shown in Figure 1, loss of body weight was evident at 8 weeks in Winnie $\times$ *Ccr6*$^{-/-}$ compared to WT. During late disease stage, the body weight was significantly increased in *Ccr6*$^{-/-}$ and Winnie $\times$ *Ccr6*$^{-/-}$ compared to the WT. The colon weight to body weight ratio was calculated at 8 weeks (WT: $0.006 \pm 0.001$, *Ccr6*$^{-/-}$: $0.007 \pm 0.001$, Winnie $\times$ *Ccr6*$^{-/-}$: $0.017 \pm 0.005$, Winnie: $0.018 \pm 0.002$) and at 16–22 weeks (WT: $0.007 \pm 0.001$, *Ccr6*$^{-/-}$: $0.026 \pm 0.003$, Winnie $\times$ *Ccr6*$^{-/-}$: $0.023 \pm 0.005$, Winnie: $0.03 \pm 0.004$). The colon weight/body weight ratio was found to be significantly increased ($p < 0.0001$(****)) in Winnie and Winnie $\times$ *Ccr6*$^{-/-}$ compared to the WT at both time points as shown in Figure 1.

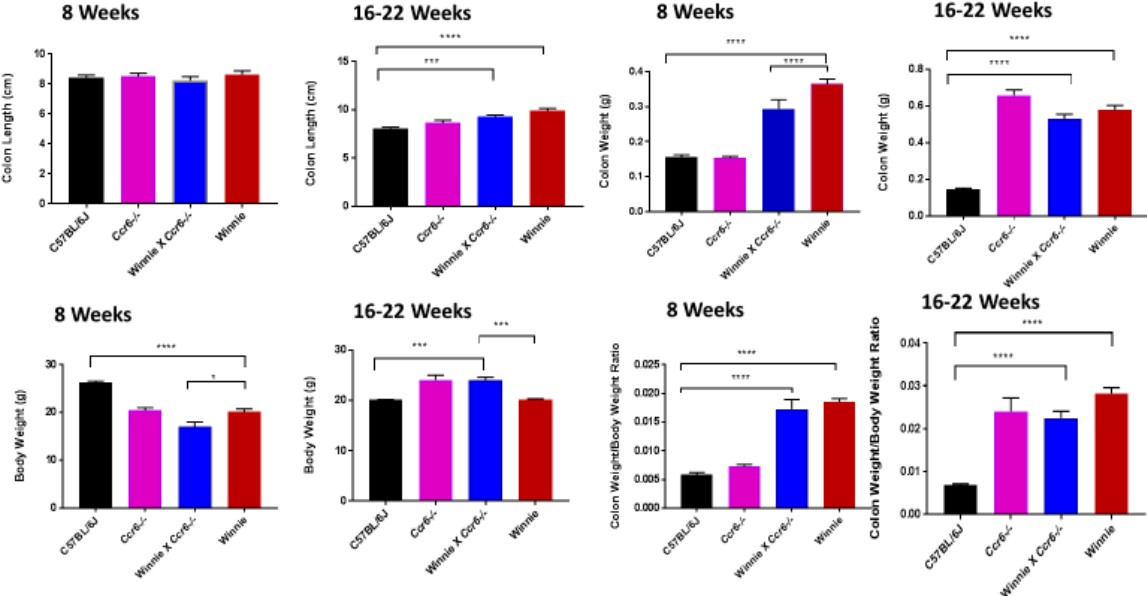

**Figure 1.** Comparison of the clinical parameters (colon length, colon weight, body weight, colon weight/body weight ratio) of freshly removed colons measured from the ileo-caecal junction to the anus in the four genotypes at 8 weeks and 16–22 weeks of age. Data expressed as Mean ± SEM by one-way analysis of variance (ANOVA) and Tukey's multiple comparison test, $n = 8$ per group. $\leq 0.001$ (***) and $p \leq 0.0001$ (****).

### 3.1.2. Disease Activity Index

The disease activity index (DAI) in the four genotypes was assessed as per the criteria given in Table 2.

**Table 2.** Summary of the disease activity index in the four genotypes from Day 1 to Day 22.

| Genotype | Body Weight | Diarrhoea | Bloody Faeces | Rectal Prolapse |
|---|---|---|---|---|
| **C57BL/6J (WT)** | Normal (25 g) | Absent | Absent | Absent |
| ***Ccr6*$^{-/-}$** | Loss of body weight | Absent | Absent | Absent |
| **Winnie $\times$ *Ccr6*$^{-/-}$** | Loss of body weight | Absent | Absent | Absent |
| **Winnie** | Loss of body weight | Present | Absent | Present |

### 3.1.3. Gross Colon Morphology

The observations related to gross colonic features made at 22 weeks after sacrifice, indicated diminished inflammation in Winnie $\times$ *Ccr6*$^{-/-}$ compared to Winnie (Table 3, Figure 2).

**Table 3.** Observations made of gross colonic morphological features after removal of colon.

| Genotype | Observation |
| --- | --- |
| **C57BL/6J (WT)** | Faecal pellets were well-formed, solid and hard with no symptoms of inflammation, narrow short colon with thin bowel wall (Figure 2A,E) |
| *Ccr6$^{-/-}$* | Semi-solid faeces in the proximal colon, faecal pellets formed in the distal colon, mild inflammation, slightly oedematous bowel wall (Figure 2B,F) |
| **Winnie × *Ccr6$^{-/-}$*** | Semi-solid faeces in the proximal colon, faecal pellets formed in distal colon, mild inflammation, mildly thickened bowel wall (Figure 2C,G) |
| **Winnie** | Watery stools, faecal pellets not formed, high inflammation, thickened bowel wall, reddened and oedematous bowel wall (Figure 2D,H) |

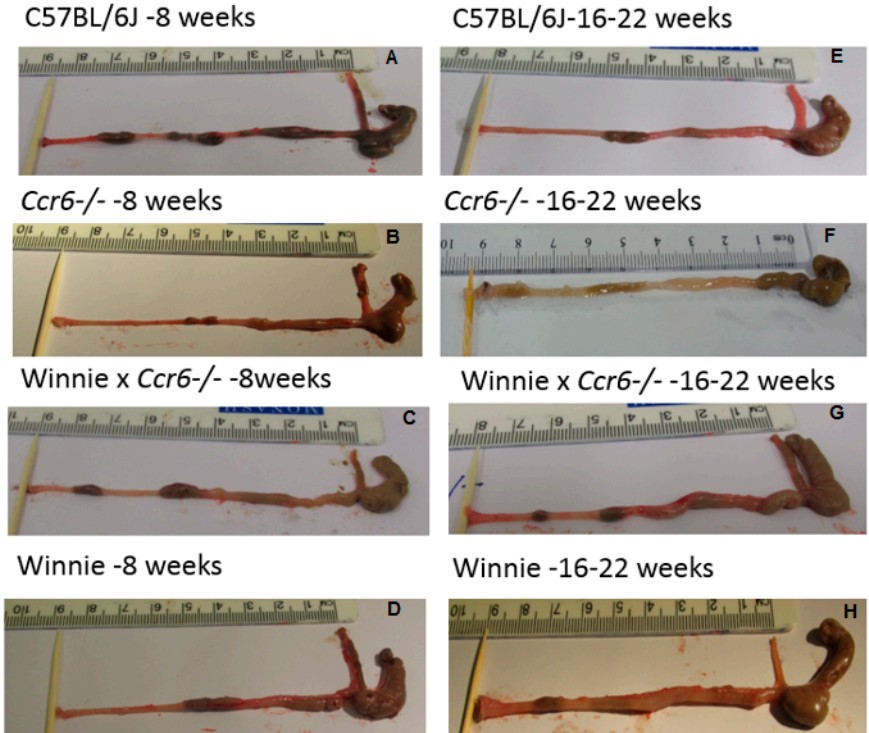

**Figure 2.** Of gross colon morphology in freshly removed colons measured from the ileo-caecal junction to the anus and before cutting open along the longitudinal axis and luminal contents removed in the four genotypes (WT, *Ccr6$^{-/-}$*, Winnie × *Ccr6$^{-/-}$* and Winnie) at 8 weeks (**A–D**) and 16–22 weeks (**E–H**) of age. Magnification ×1.

*3.2. Ccr6-Deficiency Displayed Attenuated Inflammationin Multi-System Pathology Concomitant with Colitis*

### 3.2.1. Colon Histology

Histological staining of colon tissue with H&E as shown in Figure 3, revealed normal tissue architecture with no inflammation in WT and *Ccr6$^{-/-}$* at 16–22 weeks of age. Winnie × *Ccr6$^{-/-}$* displayed histological evidence of having less severe inflammation at both time points compared to Winnie, consistent with the fact that milder inflammation was seen in the proximal colon while the distal colon was free of inflammation. Attenuated colitis initiated by *Ccr6*-deficiency was indicated in Winnie × *Ccr6$^{-/-}$*, showed increased goblet cell distribution with high mucus production compared to Winnie. *Ccr6$^{-/-}$* deficiency appeared to stimulate more epithelial mucus secretion which had contributed to reduce inflammation. Severe inflammation seen in Winnie at 16–22 weeks was not observed in Winnie × *Ccr6$^{-/-}$* to that extent. Except for moderate immune cell infiltration and an occasional crypt abscess, Winnie × *Ccr6$^{-/-}$* did not show deformed crypt architecture whereas prominent crypt abscesses, sporadic mucosal ulceration, epithelial hyperplasia and loss of goblet cells were observed in Winnie.

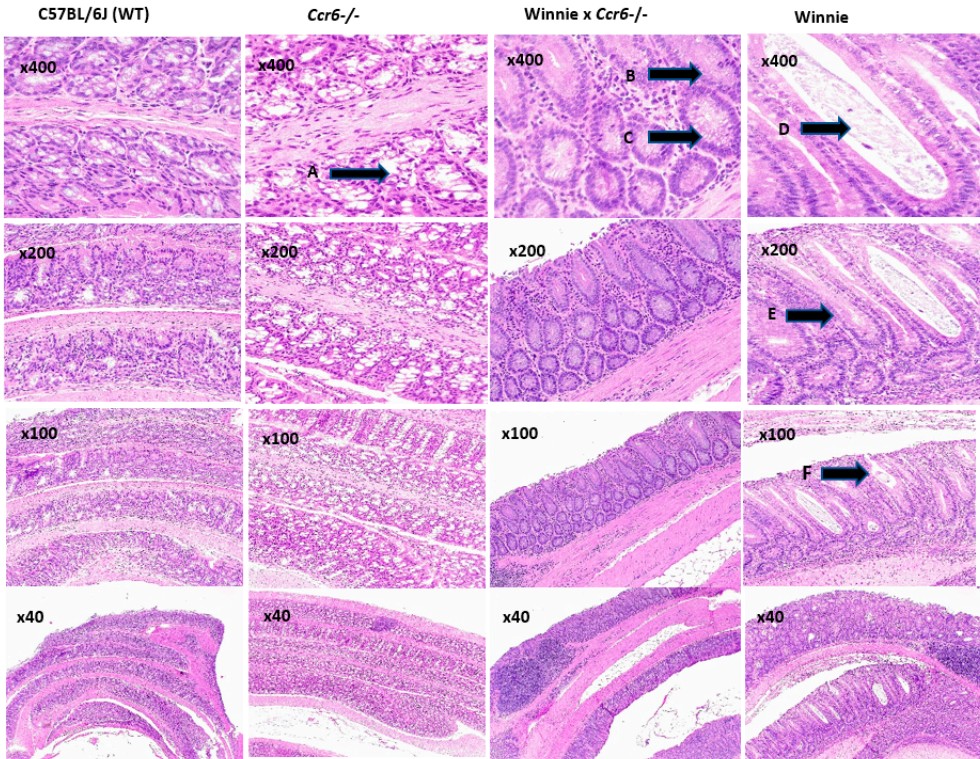

**Figure 3.** Representative images of H&E colon histomorphologyin C57BL/6J, *Ccr6*$^{-/-}$, Winnie × *Ccr6*$^{-/-}$, and Winnie at 16–22 weeks of age. Arrows indicate (**A**) mucus filled goblet cells inside short crypts in *Ccr6*$^{-/-}$, (**B**) moderately elongated crypts in Winnie × *Ccr6*$^{-/-}$, (**C**) mucus filled goblet cells in Winnie × *Ccr6*$^{-/-}$, (**D**) crypt abscess in Winnie, (**E**) highly elongated crypts in Winnie, (**F**) mucosal ulcerations in Winnie. Inflammation in the colon was evaluated using histological scheme where a score of 0–5 was used. No inflammation in WT and *Ccr6*$^{-/-}$ (histological score = 0) while moderate inflammation existed in Winnie × *Ccr6*$^{-/-}$ (histological score = 2) Severe inflammation was evident in Winnie (histological score = 5). Magnification ×40, ×100, ×200 and ×400.

### 3.2.2. Spleen Weight

No significant difference existed in the mean spleen weight among the four genotypes at 8 weeks (WT: 0.07 g ± 0.004, *Ccr6*$^{-/-}$ 0.066 g ± 0.002, Winnie × *Ccr6*$^{-/-}$ 0.082 g ± 0.008, Winnie: 0.088 g ± 0.014). A significant difference (** $p < 0.01$) was noted at 16–20 weeks of age (WT 0.09 g ± 0.007, *Ccr6*$^{-/-}$ 0.11 g ± 0.002, Winnie × *Ccr6*$^{-/-}$ 0.12 g ± 0.008 and Winnie 0.16 g ± 0.024) between WT and Winnie. Winnie × *Ccr6*$^{-/-}$ recorded a mean spleen weight intermediate to the WT and Winnie, during the second time point (Figure 4G,H). $p \leq 0.01$ (**), $p \leq 0.001$ (***) and $p \leq 0.0001$ (****).

### 3.2.3. Spleen Histology

At 8 weeks of age, the spleen in the WT and *Ccr6*$^{-/-}$ appeared normal. A considerably high neutrophil reaction was evident in the red pulp in Winnie which indicated high inflammation. In contrast, Winnie × *Ccr6*$^{-/-}$ displayed a markedly lower neutrophil reaction in the red pulp with occasional neutrophil presence. At 16–22 weeks neutrophil infiltration in the spleen was significantly higher in Winnie with high extra medullary hemopoiesis reaction (Figure 4). A similar but lower than Winnie, high neutrophil reaction was also noticeable in Winnie × *Ccr6*$^{-/-}$ at the latter time point. An increased number of megakaryocytes were observed in Winnie spleen. The spleen consists of two distinctive areas named as white pulp and red pulp. White pulp is material which forms part of the immune system (lymphatic tissue) mainly made up of white blood cells. Red pulp is made up of blood-filled cavities (venous sinuses) and splenic cords [22].

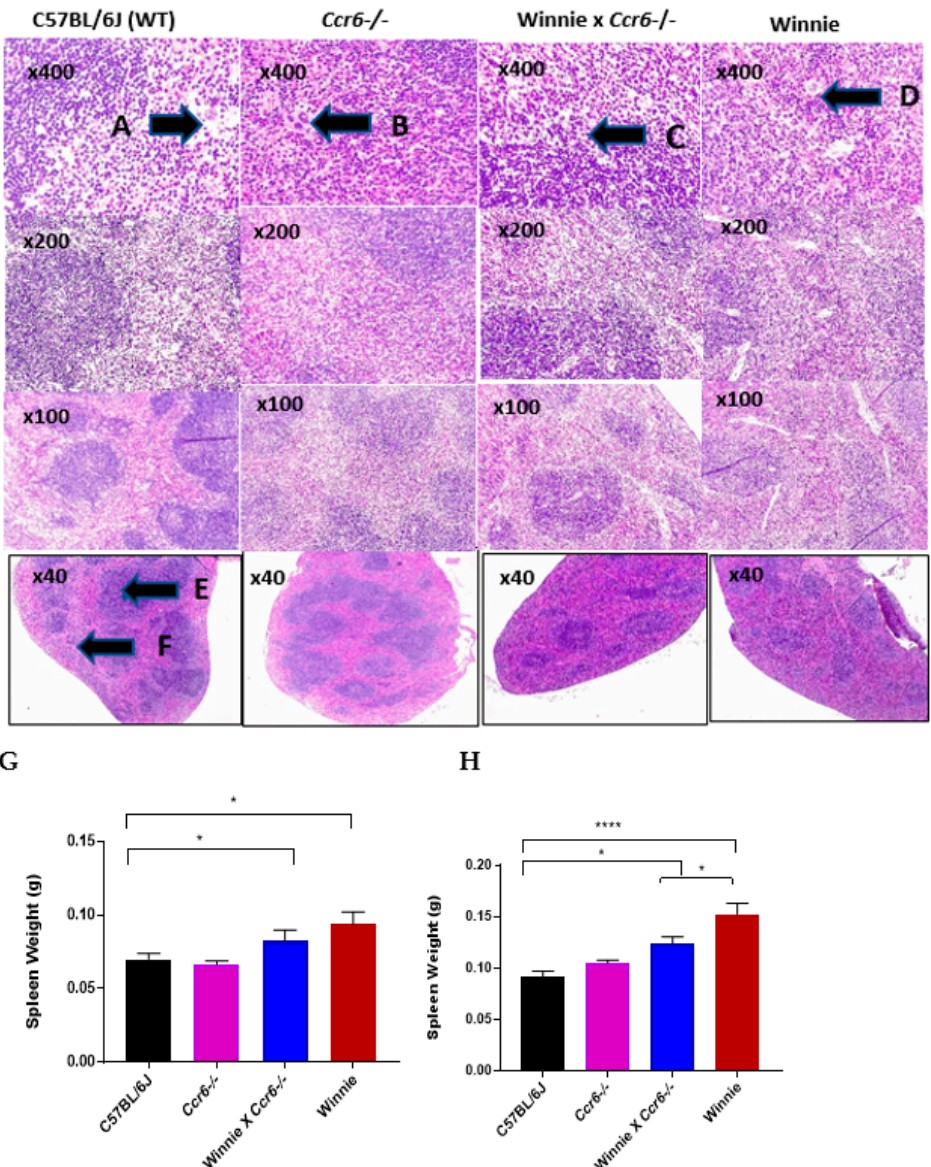

**Figure 4.** Representative images of H&E in spleen histology in the four genotypes at 16–22 weeks of age. Arrows indicate (**A**) megakaryocyte in WT, (**B**) neutrophil reaction in *Ccr6*$^{-/-}$, (**C**) neutrophil reaction in Winnie × *Ccr6*$^{-/-}$, (**D**) neutrophil reaction in Winnie, (**E**) red pulp, (**F**) white pulp, (**G**) comparison of spleen weight in the four genotypes at 8 weeks of age and (**H**) comparison of spleen weight in the four genotypes at 16–22 weeks of age. Data expressed as Mean ± SEM by one-way analysis of variance (ANOVA) and Tukey's multiple comparison test, *n* = 7, *p* ≤ 0.0001 (****). Inflammation was quantified in the genotypes by counting the numbers of neutrophils in three consecutive similar areas on the slides and comparing the mean value using a scoring index from 0–3, from no inflammation to severe neutrophil infiltration, where WT was zero, *Ccr6*$^{-/-}$ was 1, Winnie × *Ccr6*$^{-/-}$ was a moderate 2 and Winnie had a score of 3. Magnification at ×40, ×100, ×200 and ×400.

### 3.2.4. Liver Weight

The mean liver weight in the four genotypes, showed no significant difference at 8 weeks with the *Ccr6*-deficient model having the lowest mean weight (WT 1.275 ± 0.04 g, *Ccr6*$^{-/-}$ 1.114 ± 0.07 g, Winnie × *Ccr6*$^{-/-}$ 0.962 ± 0.11 g, Winnie 1.209 ± 0.12 g). At 16-20 weeks, the mean liver weights were, WT 0.98 ± 0.04 g, *Ccr6*$^{-/-}$ 1.17 ± 0.05 g, Winnie × *Ccr6*$^{-/-}$ 1.48 ± 0.07 g, Winnie 1.54 ± 0.07 g while the mean liver weight of Winnie × *Ccr6*$^{-/-}$ was lower than that of Winnie (Figure 5).

### 3.2.5. Liver Histology

No inflammation was detected in the liver of the WT at 16–22 weeks however, a large, multiple collections of inflammatory cells were observed in hepatic lobules in Winnie indicative of high focal lobular inflammation. Comparatively, $Ccr6^{-/-}$ mice, had almost none or mild infiltration of immune cells in hepatic lobules but not in the portal tracts. A smaller collection of mononuclear cells was present within the hepatic lobules in Winnie $\times$ $Ccr6^{-/-}$ identified as occasional neutrophils, eosinophils and other cells of lymphoid origin as evidenced in Figure 5.

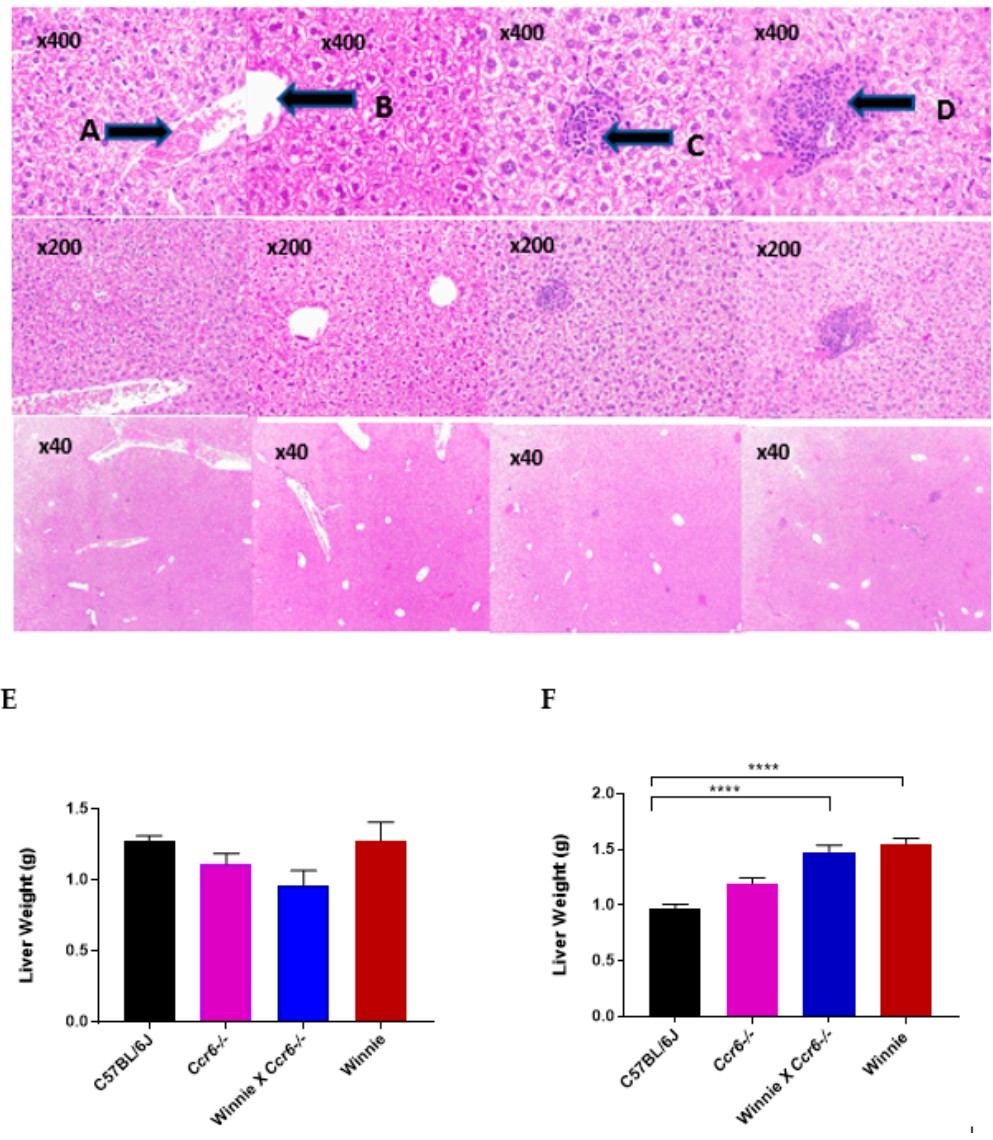

**Figure 5.** Representative images of H&E in liver histology in the four genotypes at 16–22 weeks of age. Arrows indicate (**A**) portal tract, (**B**) central canal, (**C**) immune cell infiltration indicative of focal lobular inflammation in Winnie $\times$ $Ccr6^{-/-}$, (**D**) severe focal lobular inflammation in Winnie, (**E**) comparison of liver weight in the four genotypes at 8 weeks of age, (**F**) comparison of spleen weight in the four genotypes at 16–22 weeks of age. Data expressed as Mean ±SEM by one-way analysis of variance (ANOVA) and Tukey's multiple comparison test, $n = 7$ per group. $p \leq 0.0001$ (****). Inflammation was quantified in the genotypes by counting the numbers of immune cell infiltrations which indicate sporadic focal lobular inflammation in three similar areas on the slides and comparing the mean value using a scoring index from 0–3, from no inflammation to severe neutrophil infiltration, where WT was zero, $Ccr6^{-/-}$ was 1, Winnie $\times$ $Ccr6^{-/-}$ was a moderate 2 and Winnie had a score of 3.

### 3.2.6. Kidney Weight

The mean renal weight (of both kidneys) at 8 weeks exhibited a significant difference among the four genotypes; WT: 0.310 ± 0.011 g, *Ccr6*$^{-/-}$: 0.294 ±0.017 g, Winnie × *Ccr6*$^{-/-}$: 0.275 ± 0.033 g, Winnie 0.391 ± 0.013 g with the *Ccr6*-deficient model showing an intermediate value compared to WT and Winnie. At the 16–22 weeks, (WT: 0.28 ± 0.017 g, *Ccr6*$^{-/-}$: 0.33 ± 0.019 g, Winnie × *Ccr6*$^{-/-}$: 0.36 ± 0.021 g, Winnie: 0.43 ± 0.041 g) a significant difference ($p < 0.01$ (****)) in renal weight existed between WT and Winnie (Figure 6).

### 3.2.7. Kidney Histology

As shown in Figure 6, no renal pathology was detectable in the kidneys of the four genotypes, all of which showed normal renal histology. No signs of immune cell infiltration or tubular dilatation in the kidneys were detected by histology.

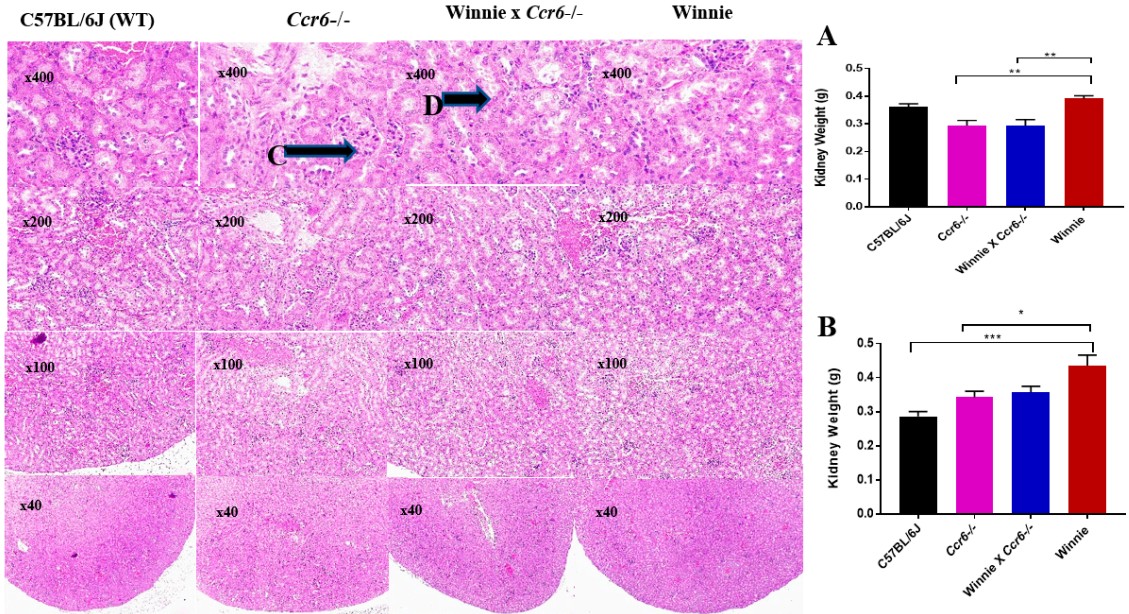

**Figure 6.** Representative images of H&E in kidney histology in the four genotypes at 16–22 weeks of age. Arrows indicate (**A**) comparison of kidney weight at 8 weeks of age (**B**) comparison of kidney weight at 16–22 weeks of age, (**C**) glomerulus (**D**) cross section of nephrons. Data expressed as mean ± SEM by one-way analysis of variance (ANOVA) and Tukey's multiple comparison test, $n = 7$ per group. $p \leq 0.01$ (**), $p \leq 0.001$ (***) and inflammation was quantified in the genotypes by counting the numbers of immune cell infiltrations in three similar repetitive areas of the samples which indicate inflammation in the kidneys.

### 3.3. Ccr6-Deficiency Reduces the Lymphocyte Distribution in the Spleen and MLN during Colitis

### 3.3.1. Gating Strategy

The gating strategy and dot plots used to quantify the lymphocyte percentages in the four genotypes in the spleen is displayed in Figure 7. In order to investigate the relative distribution of T and B lymphocytes in the spleen, the percentages of the major surface markers (CD4, CD8 and CD19), in a representative sample of 1mL splenocytes and pooled samples of MLN, were quantified by flowcytometry.

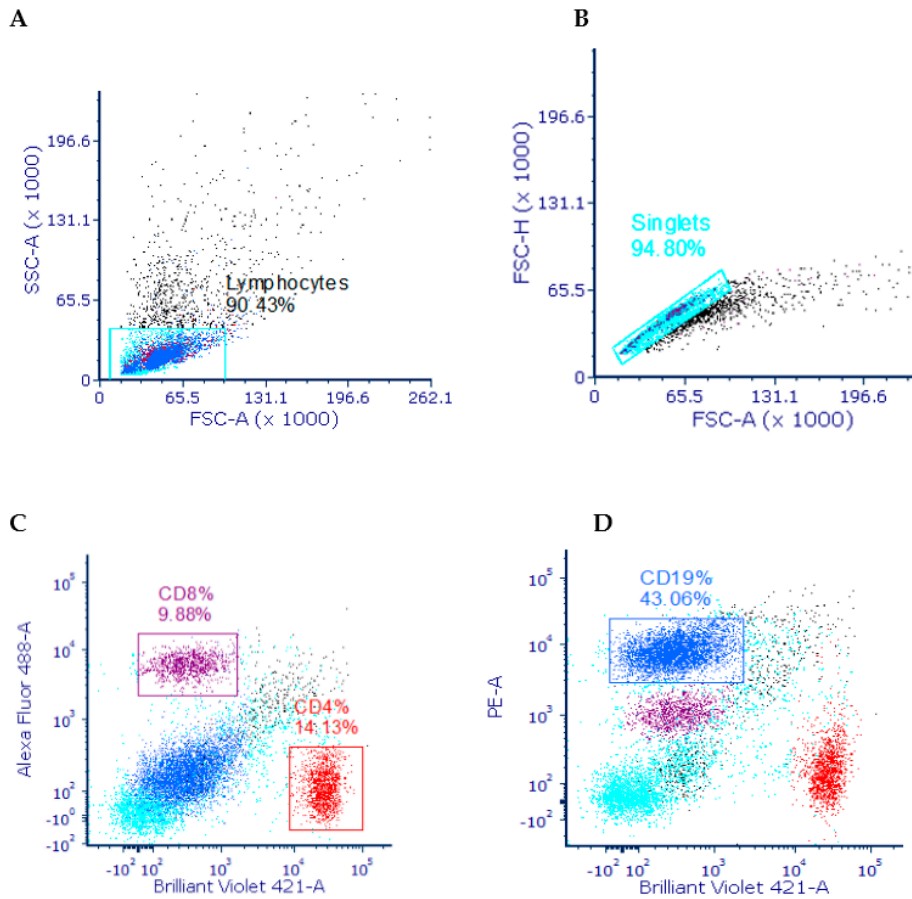

**Figure 7.** After gating the singlet cell population on FSC-H vs FSC-A axes, to discriminate against the doublets, CD4% gated in red, CD8% gated in maroon and CD19% gated in blue. Over 95% of the cell population remained viable at point of analysis. (**A**) gating for lymphocytes (**B**) separating the singlets from doublets (**C**) gating CD4 and CD8 marker = T lymphocyte percentages (**D**) gating for CD19 marker = B lymphocyte percentage.

3.3.2. T (CD4 & CD8) and B (CD19) Lymphocyte Distribution in the Spleen

As shown in Figure 8, Winnie $\times$ *Ccr6*$^{-/-}$ at both time points display reduced T and B cell production in the spleen, which may be considered a favourable outcome because lowered lymphocyte production in the spleen could result in reduced effector lymphocyte deployment from the spleen to the colon. The spleen CD4 percentage (%) at both time points show negligible difference among the four genotypes tested. At 8 weeks the mean CD4% splenic values were; WT: 12.4 $\pm$ 0.9, *Ccr6*$^{-/-}$: 13.55 $\pm$ 0.8, Winnie $\times$ *Ccr6*$^{-/-}$: 11.87 $\pm$ 0.5, Winnie: 12.81 $\pm$ 1.5 and at 16–22 weeks, WT: 10.3 $\pm$ 2.6, *Ccr6*$^{-/-}$: 10.54 $\pm$ 1.8, Winnie $\times$ *Ccr6*$^{-/-}$: 8.34 $\pm$ 2.8, Winnie: 7.0 $\pm$ 3.9. A similar trend of declining CD4% values is seen in the diseased models compared to WT at both time points. CD8% in the spleen too is non-significant among the genotypes at 8 weeks (WT: 7.93 $\pm$ 0.76, *Ccr6*$^{-/-}$: 8.62 $\pm$ 0.61, Winnie $\times$ *Ccr6*$^{-/-}$: 6.83 $\pm$ 0.91, Winnie: 8.15 $\pm$1.03) and at 16–22 weeks were, WT: 6.0 $\pm$ 0.48, *Ccr6*$^{-/-}$: 6.19 $\pm$ 0.24, Winnie $\times$ *Ccr6*$^{-/-}$: 3.61 $\pm$ 0.51, Winnie 4.35 $\pm$ 0.75. Splenic CD19% at 8 weeks were WT: 37.87 $\pm$ 3.4, *Ccr6*$^{-/-}$: 47.33 $\pm$ 2.0, Winnie $\times$ *Ccr6*$^{-/-}$: 43.51 $\pm$ 2.8, Winnie: 33.75 $\pm$ 2.7 and during the 16–22 weeks WT: 35.87 $\pm$ 1.9, *Ccr6*$^{-/-}$: 51.72 $\pm$ 1.07, Winnie $\times$ *Ccr6*$^{-/-}$: 26.22 $\pm$ 2.4, Winnie: 28.0 $\pm$ 2.9.

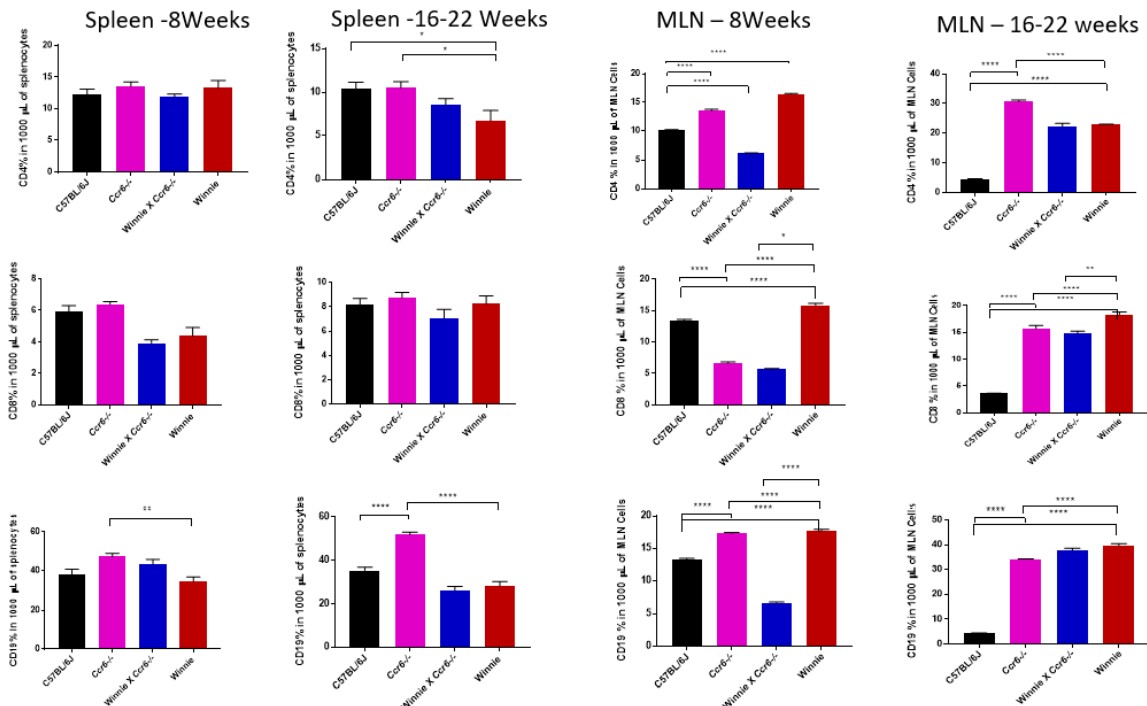

**Figure 8.** Comparison of CD4, CD8 and CD19 distribution in the spleen and MLN in the four genotypes at 8 weeks of age and 16–22 weeks of age by flowcytometry. No significant difference seen among the genotypes in the CD4% and CD8 lymphocyte % in the spleen with an increase in CD19% in the *Ccr6*-deficient models at 8 weeks although significant differences existed in the surface markers in the MLN. A trend of marked decline in all the surface marker percentages tested was seen in Winnie × *Ccr6*$^{-/-}$, also as intermediate to that of the healthy control and Winnie in both the spleen and MLN. *n* = 7 per group. Data expressed as Mean ± SEM by one-way analysis of variance (ANOVA) and Tukey's multiple comparison test. ** *p* < 0.01, **** *p* < 0.0001.

### 3.3.3. T (CD4 & CD8) and B (CD19) Lymphocyte Distribution in the Mesenteric Lymph Nodes (MLN)

Due to markedly reduced inflammation, the T and B lymphocytes in MLN of Winnie × *Ccr6*$^{-/-}$ displays decreased production during both time points compared to WT and Winnie. The CD4% in MLN lymphocyte distribution at 8 weeks; WT: 10.2 ± 0.08, *Ccr6*$^{-/-}$: 13.93 ± 0.1, Winnie × *Ccr6*$^{-/-}$: 6.26 ± 0.27, Winnie: 16.62 ± 0.16 and at 16–22 weeks; WT: 4.66 ± 0.04, *Ccr6*$^{-/-}$: 30.77 ± 0.51, Winnie × *Ccr6*$^{-/-}$: 22.19 ± 1.27, Winnie: 22.94 ± 0.07 (Figure 8). CD8 % at 8 weeks; WT: 13.5 ± 0.47, *Ccr6*$^{-/-}$: 6.93 ± 0.08, Winnie × *Ccr6*$^{-/-}$: 5.64 ± 0.08, Winnie: 15.8 ± 0.15 and at 16–22 weeks; WT: 3.65 ± 0.03, *Ccr6*$^{-/-}$: 15.74 ± 0.65, Winnie × *Ccr6*$^{-/-}$: 14.87± 0.55, Winnie: 18.38 ± 0.72. CD19 % at 8 weeks; WT: 13.0 ± 0.29, *Ccr6*$^{-/-}$: 17.0 ± 0.35, Winnie × *Ccr6*$^{-/-}$: 6.47 ± 0.24, Winnie: 17.2 ± 0.3, and at 16–22 weeks; WT: 4.31 ± 0.13, *Ccr6*$^{-/-}$: 34.18 ± 0.12, Winnie × *Ccr6*$^{-/-}$: 37.69 ± 1.1, Winnie: 39.65 ± 0.76 (Figure 8).

### 3.4. Ccr6-Deficiency Downregulates Molecular Signalling by CCL20 and PI3K$^P$, Upregulates Akt$^P$

#### 3.4.1. CCL20 Expression Pattern in the Colon

A qualitative analysis of CCL20 expression pattern in the proximal colon was made to identify the effect of the CCR6-CCL20 axis on the gut immune mechanisms during spontaneous colitis using IHC [23]. CCL20 expression pattern at 16–22 weeks of age in the four genotypes were WT: 0.006 ± 0.0003, *Ccr6*$^{-/-}$: 0.012±0.0003, Winnie × *Ccr6*$^{-/-}$: 0.012 ± 0.0003, Winnie: 0.015 ± 0.0003. CCL20 expression was highest in Winnie while *Ccr6*-deficienct models exhibited a comparatively diminished CCL20 expression to Winnie (Figure 9A–D).

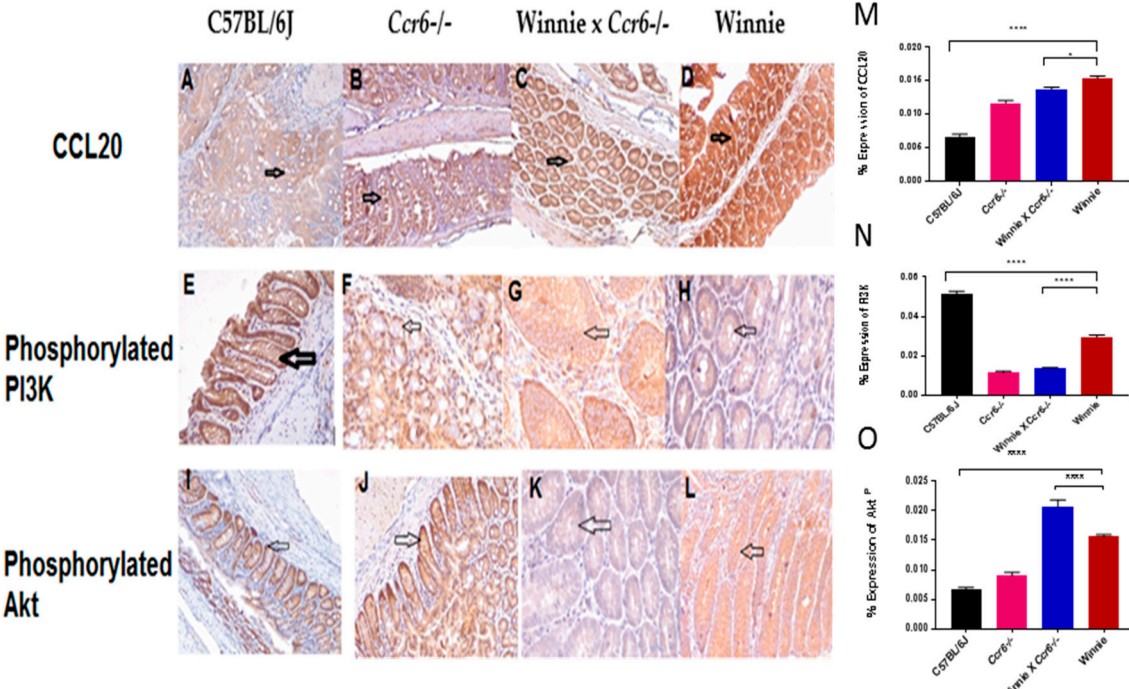

**Figure 9.** Representative images of the CCL20 (**A–D**), PI3K$^P$ (**E–H**) and phosphorylated Akt (**I–L**) expression in the colonic epithelium in the four genotypes at 16–22 weeks by IHC. Arrow indicates brown colour which determines the presence of DAB chromogen bound to anti-CCL20, anti-PI3K$^P$ and anti-Akt$^P$ mouse primary antibody, magnified at ×200. The % expression of CCl20 (M), PI3K$^P$(N) and phosphorylated Akt (O) quantified by Fiji image J software showing increased CCL20 and Akt$^P$ expression in the colon with a significant decrease in PI3K$^P$ expression in Winnie × *Ccr6*$^{-/-}$ compared to WT. In all three measurements made between Winnie × *Ccr6*$^{-/-}$ and Winnie, a significant reduction in these signalling molecules was detected in the *Ccr6*-deficient Winnie. *n* = 6 per group. Data expressed as Mean ± SEM by one-way analysis of variance (ANOVA) and Tukey's multiple comparison test. **** *p* < 0.0001.

### 3.4.2. Phosphorylated Phosphoinositide 3-Kinase (PI3K$^P$) Expression Pattern in the Colon

PI3K$^P$ expression pattern in the proximal colon was made to identify the molecular signalling stimulated by the CCR6-CCL20 axis during spontaneous colitis. The expression pattern of PI3K$^P$ values at 16–22 weeks of age, in the four genotypes were WT: 0.05 ±0.001, *Ccr6*$^{-/-}$: 0.011 ± 0.001, Winnie × *Ccr6*$^{-/-}$: 0.014 ± 0.0003, Winnie: 0.029 ± 0.001. It was evident that *Ccr6*-deficiency mediated a remarkable suppression of PI3K expression in the colon during colitis (Figure 9E–H). PI3K$^P$ activity is synonymous with the mobilisation of CCR6 -bearing immune cells, which in here shows that the *Ccr6*-deficient models may have significantly lowered immune cell chemotaxis in the colon (Figure 9E–H).

### 3.4.3. Phosphorylated Akt (Akt$^P$) Expression Pattern in the Colon

Phosphorylated Akt (Akt$^P$) expression pattern in the proximal colon was evaluated to identify the molecular signalling activity by the CCR6-CCL20 axis towards immune homeostasis during spontaneous colitis. The Akt$^P$ expression at 16–22 weeks in the four genotypes were WT: 0.007 ± 0.0003, *Ccr6*$^{-/-}$: 0.009 ± 0.0005, Winnie × *Ccr6*$^{-/-}$: 0.02 ± 0.001, Winnie: 0.015 ± 0.0003. Compared to WT, the *Ccr6*-deficient Winnie model showed a significantly high expression of Akt$^P$ in the colon (Figure 9I–L).

*3.5. Ccr6-Deficiency Suppress the T Helper Lymphocyte Immune Responses During Colitis and Upregulates IL-10*

The cytokine mRNA expression in colon tissue (Table 4) was quantified in the four genotypes and a decrease in the mRNA fold change expression normalised to the WT was observed in Winnie × $Ccr6^{-/-}$. In most cytokines at both time points, Winnie × $Ccr6^{-/-}$ mRNA expressions were lower than that of Winnie. CD is characterised by TH1 cytokines, particularly Interferon-gamma (IFN-γ), Tumour necrosis factor-alpha (TNF-α), Interleukin-18 (IL-18) and Interleukin-12 (IL-12). UC is considered as having a TH2-like cytokine profile including Interleukin-4 (IL-4), Interleukin-6 (IL-6) and Interleukin-13 (IL-13), all of which reduced disease severity in various mouse models and humans when blocked with antibodies of the respective TH1 and TH2 cytokine milieu [24].

**Table 4.** The cytokine expression evaluated by RT-PCR, divided into respective immune responses of the T helper lymphocyte populations.

| TH1 | TH2 | TH17 | Treg |
|-----|-----|------|------|
| IFN-γ | IL-4 | IL-17A | IL-10 |
| TNF-α | IL-6 | IL-23R | |
| IL-18 | IL-13 | | |

### 3.5.1. TH1 Cytokine mRNA Expression in the Colon

The TH1 cytokine profile included Interferon-gamma (IFN-γ), Tumour necrosis factor-alpha (TNF-α), Interleukin-18 (IL-18) mRNA expression at 8 weeks and 16–22 weeks of age. The IFN-γ values at 8 weeks were (WT: 1.0 ± 0, $Ccr6^{-/-}$: 0.7 ± 0.2, Winnie × $Ccr6^{-/-}$: 1.0 ± 0.8, Winnie: 10.0 ± 4.5) and at 16–22 weeks (WT: 1.0 ± 0, $Ccr6^{-/-}$: 2.0 ± 1.0, Winnie × $Ccr6^{-/-}$: 3.0 ± 1.0, Winnie: 3.0 ± 1.5). TNF-α values at 8 weeks were (WT: 1.0 ± 0, $Ccr6$-/-: 1.0 ± 0.45, Winnie × $Ccr6^{-/-}$: 2.0 ± 1.8, Winnie: 10 ± 2.5) and at 16–20 weeks (WT: 1.0 ± 0, $Ccr6^{-/-}$: 1.0 ± 0.3, Winnie × $Ccr6^{-/-}$: 3.0 ± 2.2, Winnie: 4.0 ± 3.6). IL-18 values at 8 weeks were (WT: 1.0 ± 0, $Ccr6^{-/-}$: 2.25 ± 1.0, Winnie × $Ccr6^{-/-}$: 10.0 ± 1.8, Winnie: 39.0 ± 1.0) and at 16–20 weeks (WT: 1 ± 0, $Ccr6^{-/-}$: 12.0 ± 5.8, Winnie × $Ccr6^{-/-}$: 5.0 ± 3.4, Winnie: 12.0 ± 5.8). Winnie × $Ccr6^{-/-}$ TH1 cytokine mRNA expression were found to be lower than that of the positive control at both time points (Figure 10).

### 3.5.2. TH2 Cytokine mRNA Expression in the Colon

The TH2 cytokine profile included Interleukin 4 (IL-4), Interleukin 6 (IL-6) and Interleukin 13 (IL-13) mRNA expression at 8 weeks and 16–22 weeks of age. Winnie × $Ccr6^{-/-}$ mRNA expression of the three cytokines were found to be either similar to that of WT or higher than that of the positive control at the second time point (Figure 10).IL-4 values at 8 weeks; WT: 1.0 ± 0, $Ccr6^{-/-}$: 0.3 ± 0.11, Winnie × $Ccr6^{-/-}$: 1.0 ± 0.8, Winnie: 35.0 + 1.0 and at 16–22 weeks; WT: 1.0 ± 0, $Ccr6^{-/-}$: 0.5 ± 0.06, Winnie × $Ccr6^{-/-}$: 5.0 ± 3.5, Winnie: 8.0 ± 5.6. IL-6 values at 8 weeks; WT: 1.0 ± 0, $Ccr6^{-/-}$: 0.4 ± 0.1, Winnie × $Ccr6^{-/-}$: 3.0 ± 2.2, Winnie: 9.0 ± 1.0, and at 16–22 weeks; WT: 1.0 ± 0, $Ccr6^{-/-}$: 1.0 ± 0.5, Winnie × $Ccr6^{-/-}$: 1.0 ± 0, Winnie: 3.0 ± 2.0. IL-13 values at 8 weeks; WT: 1.0 ± 0, $Ccr6^{-/-}$: 0.3 ± 0, Winnie × $Ccr6^{-/-}$: 2.0 ± 1.0, Winnie: 22.0 ± 1.0 and at 16–22 weeks; WT: 1.0 ± 0, $Ccr6^{-/-}$: 1.0 ± 0, Winnie × $Ccr6^{-/-}$: 4.0 ± 1.5, Winnie: 4.0 ± 1.0.

### 3.5.3. TH17 Cytokine mRNA Expression in the Colon

The TH17 cytokine profile included Interleukin 17A (IL-17A) and Interleukin 23 Receptor (IL-23R) mRNA expression at 8 weeks and 16–22 weeks of age. IL-17 A values at 8 weeks; WT: 1.0 ± 0, $Ccr6^{-/-}$: 1.0 ± 0.6, Winnie × $Ccr6^{-/-}$: 2.0 ± 0, Winnie: 35.0 ± 1.0 and at 16–22 weeks; WT: 1.0 ± 0, $Ccr6^{-/-}$: 1.0 ± 0.6, Winnie × $Ccr6^{-/-}$: 2.0 ± 0, Winnie: 35.0 ± 1.0. IL-23R values at 8 weeks; WT: 1.0 ± 0, $Ccr6^{-/-}$: 0.3 ± 0.2, Winnie × $Ccr6^{-/-}$: 1.0 ± 0.3, Winnie: 62.0 ± 1.0 and at 16–22 weeks; WT: 1.0 ± 0, $Ccr6^{-/-}$:

0.3 ± 0.15, Winnie × *Ccr6*⁻/⁻: 2.0 ± 1.8, Winnie: 2.0 ± 1.8. Winnie × *Ccr6*⁻/⁻ TH17 cytokine mRNA expressions were found to be lower than that of the positive control at both time points (Figure 10).

### 3.5.4. Anti-Inflammatory (Treg) Cytokine mRNA Expression in the Colon

The anti-inflammatory cytokine expression in the colon of the four genotypes were evaluated in IL-10 cytokine profile at 8 weeks; (WT: 1.0 ± 0, *Ccr6*⁻/⁻: 2.0 ± 1.0, Winnie × *Ccr6*⁻/⁻: 1.0 ± 1.0, Winnie: 70.0 ± 1.0) and at 16–22 weeks (WT: 1.0 ± 0, *Ccr6*⁻/⁻: 2.5 ±1.0, Winnie × *Ccr6*⁻/⁻: 4.0 ± 1.5, Winnie: 4.0 ± 1.0). Winnie × *Ccr6*⁻/⁻ showed a higher four-fold increase in IL-10 mRNA expression than Winnie at the second time point (Figure 10).

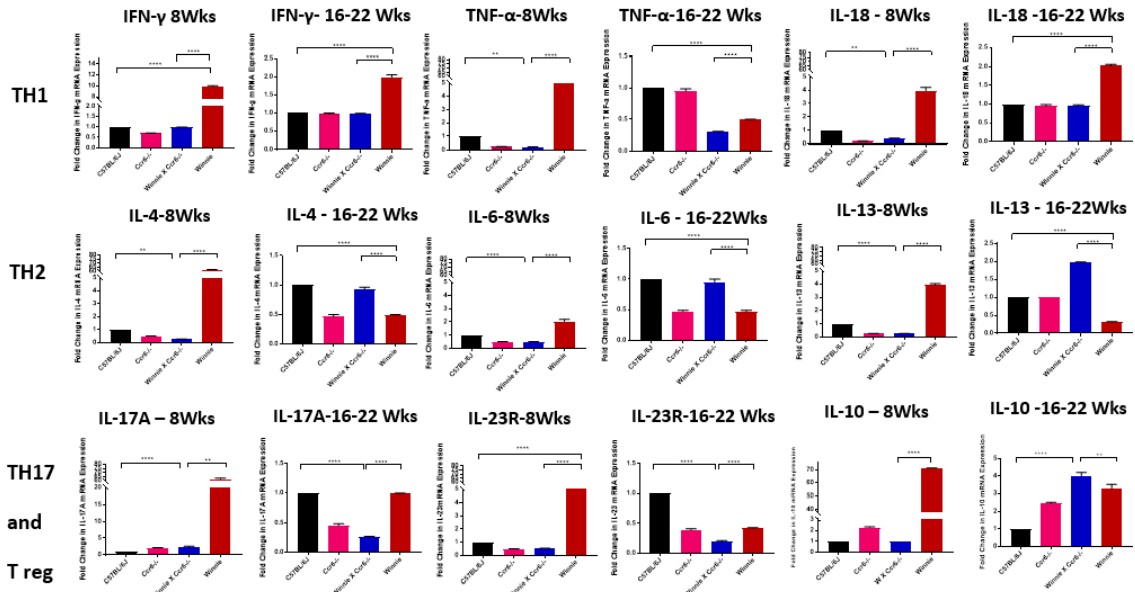

**Figure 10.** Fold change in mRNA expression of IFN-g, TNF-α, IL-18, IL-4, IL-6, IL-13, IL-17A, IL-23R, IL-10 at 8 weeks and 16–22 weeks. Cycle of threshold (CT) values were normalised to *Gapdh* and CT calculated using the method of $2^{-\Delta CT}$. Data expressed as Mean ±SEM by One- way analysis of variance (ANOVA) and Tukey's multiple comparison test, $n = 6$ per group. \*\*$p < 0.01$, \*\*\*\* $p < 0.0001$.

## 4. Discussion

The purpose of this investigation was to identify the phenotype of Winnie × *Ccr6*⁻/⁻ mouse model and evaluate *Ccr6*-deficiency in multi-organ pathophysiology and its role in molecular signalling during the manifestation of spontaneous chronic colitis. The results of this study have revealed that CCR6 plays a pathogenic role during the development of chronic colitis. As a direct result of *Ccr6*-deficiency during the continuance of colitis in Winnie, the following salient features were observed;(i) distinct reduction in colonic inflammation, spleen and liver (ii) absence of renal pathology (iii) suppressed TH1 and TH17 immune responses in the colon (iv) restrained production and egression of lymphocytes from the spleen (v) high lymphocyte production within the mesenteric lymph nodes and (vi) increased production of CCL20, and Akt$^P$ with decreased production of PI3K$^P$ in the colon.

Both human and mice systems have provided evidence for the role played by CCR6-CCL20 axis in the progression and pathogenicity of several autoimmune disorders, as well as IBD [1]. The study model used in this investigation, namely Winnie which carries a missense point mutation in the *Muc2* gene displays a similar but a relatively less severe intestinal inflammation that mimics human colitis. Winnie mouse model as a tool has multiple advantages in comparison to other contrived colitis models which include (i) chemically induced models such as DSS and TNBS induced colitis (no need for toxic chemicals) (ii) adoptive T cell transfer induced colitis (induced in an artificial immune deficient system) (iii) genetically engineered models of colitis (complete deficiency of immune functions) for

the following reasons such as (i) disease manifests as spontaneous colitis, (ii) mimics human UC, (iii) doesn't cause severe tissue damage and (iv) it maintains an intact immune system resembling a normal immune system in comparison to T cell transfer models [19]. Additionally, inflammation in Winnie naturally occurred at 6 weeks of age [17] and progressed with age. Backcrossing Winnie to *Ccr6* deficient mice created a unique system that enabled us to study the role played by *Ccr6* in an established chronic colitis setting.

The *Ccr6*-deficient Winnie model demonstrated moderate weight loss, absence of faecal occult blood, and diarrhoea with fully formed faecal pellets in the distal colon. An increase in wet colon weight and shortened colon length indicate elevated inflammation [25] due to fluid accumulation and thickening of the colon attributed to oedematous swelling and thickening of bowel wall. Colon weight to body weight ratio which is often treated as a quantitative index of colonic inflammation, when calculated, appeared to have reduced inflammation in Winnie × $Ccr6^{-/-}$ compared to Winnie. All the clinical parameters described above lead to the conclusion that *Ccr6* deficiency renders Winnie with reduced colitis. Our data is in complete agreement with results obtained by several research groups that have utilised the *Ccr6* deficient mouse systems. For example, *Ccr6*-deficient mice induced by DSS had displayed reduced susceptibility to the disease [10] while TNBS - induced colitis was attenuated by blocking the release of CCL20 [26]. The adoptive transfer of CCR6$^+$ CD4 T cells into recipient recombination activation gene 2 ($Rag2^{-/-}$) mice also caused aggravated disease [21].

The CCR6-CCL20 axis plays a pivotal role in maintaining immune homeostasis in the gut [24]. It produces immune tolerance against the plethora of infectious luminal microbes which penetrate the gut mucosa and simultaneously step up immunity against many other disease-causing secondary contributing factors. When the purported immune tolerance breaks down, the phenomenal TH17 Vs Treg imbalance paradigm comes into play either by increased multiplication of pro-inflammatory immune cells such as CCR6$^+$ TH17 and CCR6$^+$ TH1, or by immune mechanisms that suppress the production of Treg cells [21]. This scenario is empowered by CCR6 which mainly functions in the role of directing immune cell chemotaxis and thus stimulating a cascade of accessory molecules and cytokines.

The attenuation in inflammation was consistently evident in the colonic histomorphology (with increased goblet cell numbers which stimulated an increased production of mucus) seen in the study model in comparison to Winnie (which has deficient mucus generation). Features of acute disease histology were less prominent in Winnie × $Ccr6^{-/-}$ with all the general criteria used in the assessment such as epithelial hyperplasia, crypt abscesses, mucosal ulceration and immune cell infiltration being markedly reduced, showing a milder inflammatory status which correlated well with the clinical parameters.

In addition to having pathology within the gastrointestinal tract, IBD patients also exhibit secondary organ pathologies termed as extraintestinal manifestations [12]. The spleen, liver and kidneys which are immune-related organs of secondary lymphoid origin were examined as they cause the systemic diseases—hepatitis and glomerular nephritis, where CCR6 plays a pivotal role in initiating inflammation [13–16]. Renal histology remained normal and unaffected by the *Ccr6*-gene ablation although there have been reports on *Ccr6*-deficiency having aggravated renal injury and increased mortality among nephritic mice because compared with the WT, *Ccr6*-deficiency in mice had reduced infiltration of regulatory T cells (Treg cells) and TH17 cells but not the TH1 type [27].

In contrast to displaying normal kidney histology, liver pathology indicated significant lobular inflammation in Winnie with milder inflammation in the *Ccr6*-deficient Winnie [28]. There are reports which describe the occurrence of concomitant hepatobiliary manifestations in IBD patients, of which the most common complication being primary sclerosing cholangitis (PSC) [29]. Research findings have evaluated the presence of liver extrahepatic manifestations associated with IBD [30]. Consistent with the mild inflammation seen in this study, there were no granulomas, hepatic abscesses, amyloidosis, and gallstones which are traditionally observed in CD, while PSC and autoimmune hepatitis are usually described in UC patients [29]. One of the important observations made in this investigation is that *Ccr6*-deficiency serves to reduce the disease severity in the liver and spleen of colitis-afflicted mice.

The distribution of the major lymphocyte types in the spleen were ascertained with respect to *Ccr6* gene deficiency in order to highlight whether an immune-specific deficiency in the lymphocyte arm is created. CD4, CD8 and CD19 are major surface markers of the T and B lymphocyte repertoire in the spleen. Heightened inflammation in Winnie could be attributed to an imbalance in immune responses of T and B lymphocytes produced in the spleen because they are less in number. The impact of *Ccr6*-deficiency on lymphocyte production in the spleen produces suppressed effector cell mobilization and dysfunctional lymphocyte egression from the spleen. FTY720, an inhibitor of the sphingosine 1 phosphate receptor (SIP1) agonist has been shown to restrict CD4 T lymphocyte egression from lymph nodes during the disease course of allergic diarrhoea [31]. *Ccr6*-deficiency in Winnie may possibly produce similar SIP1 receptor inhibition reactivity resulting in diminished CD4 lymphocyte egression from the spleen in Winnie during acute colitis, which needs to be investigated further.

IFN-γ, TNF-α and IL-18 are considered as pro-inflammatory cytokines in IBD. Published data have revealed a robust production of IFN-γ in the gut of DSS-treated WT mice associated with severe intestinal inflammation [32]. Neutralization antibody against IFN-γ had partially but significantly ameliorated disease [33]. IFN-γ which is regarded as a key TH1- upregulated - inflammatory cytokine, is also involved in regulating intestinal epithelial cell homeostasis, cell proliferation and apoptosis through converging beta- catenin signalling pathways [34], and activating the major histocompatibility complex class II (MHC II) on antigen-presenting cells and non-immune cells [35]. The flip side of IFN-γ being a key inflammatory cytokine is that it bears anti-inflammatory properties too which may be supported by the observations made at the latter time point. It was shown in a mouse model of *Salmonella typhimurium*-induced colitis, that in animals lacking IFN-γ, the severity of intestinal inflammation was markedly attenuated [36]. Moreover, IFN-γ has been identified as a negative regulator of IL-23-mediated experimental colitis [37].

High levels of TNF-α in the intestinal mucosa is associated with the development of relapses and sustaining chronic inflammatory activity, which in this study model was markedly reduced. Intestinal homing of memory T lymphocytes were shown to augment TNF-α levels in relapsing UC patients along with higher NF-κB nuclear translocation in lamina propria mononuclear cells taken from IBD patients [38].

The TH2 cells produce IL-4, IL-5, IL-6, IL-10 and IL-13 and promote atopy through activation of mast cells and induction of Ig E immune responses. IL-4, IL-10 and IL-13 are known to inhibit TH1-mediated immune responses [39] which have been evident in *Ccr6*-deficient Winnie.

In this study, *Ccr6*-deficiency significantly induced a remarkably high IL-10 expression at16–22 weeks, during latedisease stage. This may be one contributing factor which had favoured reduced inflammation shown by Winnie × *Ccr6*$^{-/-}$ at the second time point. IL-10 is an established anti-inflammatory cytokine produced by T cells, B cells and monocytes when stimulated by an antigenic stimulus. IL-10 is known to downregulate MHC II molecules which diminish the antigen presenting capacity of cells thus inhibiting the production of IL-6, IL-1β and TNF-α [39]. IL-17 and IL-23 are potent inflammatory cytokines which induce opposing effects in colitis. Inhibition of IL-17 had exacerbated CD and weakened epithelial barrier integrity, thus exhibiting its potency for activating disease resolution. IL-23 inhibition had attenuated CD as well as promoting the development of the regulatory T cells [40]. In another study, antibodies targeting IL-23 had ameliorated colitis [41]. Results of this study have shed some light towards being supportive of these observations because both IL-17A and IL-23 appear to perform dual roles in colitis.

The presence of phosphoinositide 3-kinase (PI3K) in the healthy control is due to PI3K activation by G-protein coupled receptor repertoire [42]. PI3K is associated with a host of cellular activities including cell cycle, cell survival, cell proliferation etc. PI3K in Winnie and Winnie × *Ccr6*$^{-/-}$ models confirm that the AKT/mTOR pathway, NF-κB pathway, MAPK pathway, are possibly upregulated inducing inflammation in the murine colon during spontaneous colitis [43]. The PI3K pathway, has a critical signal transduction system linking oncogenes and multiple receptor classes to many essential cellular functions. It is perhaps the most commonly activated signalling pathway in human

cancer [44]. A remarkable fact is that the *Ccr6*-deficiency in Winnie had downregulated the PI3K expression in the colonic epithelial tissue. Winnie × *Ccr6*$^{-/-}$ was shown to have the highest percentage expression in phosphorylated Akt, during late disease, from which we could possibly deduce it to behave as a favourable mediator which had contributed to reduce inflammation in murine colitis. Protein kinase B (PKB), also known as Akt, is a serine/threonine-specific protein kinase that plays a key role in multiple cellular processes such as glucose metabolism, apoptosis, cell proliferation, transcription and cell migration [45]. Among the network of immune pathways associated with Akt, the CCR6-stimulated Akt/mTOR signalling pathway leads to the activation of STAT3 which decisively selects the differentiation of T cells into TH17 and Treg populations in the gut [21]. The spectrum of biological functions thus initiated by phosphorylated Akt in both pre-clinical and clinical models is wide and varied. Evidence-based research studies have elucidated several immunological pathways beginning with the activation of CCR6 leading towards upregulating physiological mechanisms in addition to promoting the ERK, MAPK and NF-κB pathways [46].

The limitations of the study include lack of functional studies on the immune cell repertoires and quantification of regulatory T cells and other T lymphocyte cohorts as well as detailed screening of the proteins involved in molecular signalling which will be the next step in our research study.

Recent GWAS studies illustrated *Ccr6* as a predisposing genetic locus in IBD patients, a mechanism by which CCR6 imparts immunological modulation [47]. This study has highlighted a functional role for CCR6 in the pathogenicity of spontaneous chronic colitis and provides evidence that CCR6 could be utilised as a functional therapeutic target in IBD. The proven functional efficacy in CCR6-inhibition in psoriasis, rheumatoid arthritis, multiple sclerosis carcinoma and EAE, could be replicated in IBD as well [11]. Recent research had shown the direct involvement of CCR6 with AKT/mTOR pathway which contributes to the TH17/Treg imbalance which is considered as the prime immunological factor which decides the resolution of colitis [21] and needs further investigation. In conclusion, it is proposed to further evaluate the potency of CCR6-inhibition as an active drug target in the clinical studies of colitis.

Conclusion: *Ccr6*-deficiency appears to have reduced the severity of colitis in the Winnie mouse model.

**Author Contributions:** Conceptualization, R.R.; Data curation, A.P.P., W.B. and P.S.; Formal analysis, R.F.; Supervision, M.S., T.P. and R.E. All authors have read and agreed to the published version of the manuscript.

**Acknowledgments:** The first author is funded by a research training programme (RTP) scholarship from the Australian government.

**Conflicts of Interest:** Authors declare no conflict of interests, commercial or financial in writing this article.

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
