# Peer review of "Ccr6 Deficiency Attenuates Spontaneous Chronic Colitis in Winnie"

_gastrointestdisord, doi:10.3390/gidisord2010004_

Round 1
Reviewer 1 Report
Inflammatory bowel diseases (IBDs) display a severe health burden to diseased individuals and are expected to rise in prevalence. Patients have to cope with several co-morbidities such as anemia, decreased bone mineral density, depression and, importantly, an increased risk to develop colorectal cancer. Therefore, it is essential to elucidate underlying molecular mechanisms in order to develop therapeutic strategies. In their study, the authors aimed to elucidate the consequences of Ccr6 deficiency in the Winnie mouse model for colitis. Their findings on this topic will be of interest to the readership of Gastrointestinal Disorders. A number of minor points should be addressed prior to publication as detailed below.
How exactly was the DAI assessed? Were diarrhea and occult blood only scored as “absent” or “present” or did the authors determine scores? For instance, the stool consistency can be scored in a range from 0-3 (normal, soft, very soft, diarrhea). Which method was applied to test for the presence of occult blood? Was it determined macroscopically or was a hemoccult test/guaiac test performed?
Please improve the presentation of the gross colon morphology (Figure 2). The scale is not clearly visible in all images.
To support their findings on the inflammation intensity (Figure 3), the authors should quantify the degree of epithelial damage by determining the H-Score/Histo-Score. This score is based on the intactness of the crypt structures as well as on the degree of the infiltration of immunoregulatory cells.
Please add scale bars to figures 3, 4, 6 and 9. Moreover, in Figure 3 the 40x magnification of the Winnie, Ccr6-/- is not consistent with the morphology shown in the images displaying the 100/200/400x magnification.
The underlying signaling pathways suggested in Figure 11 are not clearly supported by the data provided by the authors and are rather speculative. Instead of showing these pathways in a figure, it will be sufficient to mention them in the discussion section
Reviewer 2 Report
The authors, using a Ccr6 deficient mouse model, investigated pathophysiology of chronic colitis. CCR6 deficieny was shown to attenuate inflammation in the spleen, liver and colon.
The manuscript is well-written, and the conclusions are based on the results.
